# The claudin-like apicomplexan microneme protein is required for gliding motility and infectivity of *Plasmodium* sporozoites

**Manon Loubens**[1¤], **Carine Marinach**[1], **Clara-Eva Paquereau**[1], **Soumia Hamada**[2], **Bénédicte Hoareau-Coudert**[3], **David Akbar**[4], **Jean-François Franetich**[1], **Olivier Silvie** [1]*

**1** Sorbonne Université, INSERM, CNRS, Centre d'Immunologie et des Maladies Infectieuses, CIMI-Paris, Paris, France, **2** Sorbonne Université, INSERM, UMS PASS, Plateforme Post-génomique de la Pitié Salpêtrière (P3S), Paris, France, **3** Sorbonne Université, INSERM, UMS PASS, Plateforme de cytométrie de la Pitié-Salpêtrière (CyPS), Paris, France, **4** Sorbonne Université, INSERM, CNRS, Hôpital de la Pitié Salpêtrière, Paris Brain Institute, ICM Quant Cell imaging Core Facility, Paris, France

¤ Current address: Laboratory of Pathogens and Host Immunity (LPHI), Université de Montpellier, CNRS, Montpellier, France.
* olivier.silvie@inserm.fr

**Data Availability Statement:** All relevant data are within the manuscript and its Supporting Information files.

## Abstract

Invasion of host cells by apicomplexan parasites such as *Toxoplasma* and *Plasmodium* spp requires the sequential secretion of the parasite apical organelles, the micronemes and the rhoptries. The claudin-like apicomplexan microneme protein (CLAMP) is a conserved protein that plays an essential role during invasion by *Toxoplasma gondii* tachyzoites and in *Plasmodium falciparum* asexual blood stages. CLAMP is also expressed in *Plasmodium* sporozoites, the mosquito-transmitted forms of the malaria parasite, but its role in this stage is still unknown. CLAMP is essential for *Plasmodium* blood stage growth and is refractory to conventional gene deletion. To circumvent this obstacle and study the function of CLAMP in sporozoites, we used a conditional genome editing strategy based on the dimerisable Cre recombinase in the rodent malaria model parasite *P. berghei*. We successfully deleted *clamp* gene in *P. berghei* transmission stages and analyzed the functional consequences on sporozoite infectivity. In mosquitoes, sporozoite development and egress from oocysts was not affected in conditional mutants. However, invasion of the mosquito salivary glands was dramatically reduced upon deletion of *clamp* gene. In addition, CLAMP-deficient sporozoites were impaired in cell traversal and productive invasion of mammalian hepatocytes. This severe phenotype was associated with major defects in gliding motility and with reduced shedding of the sporozoite adhesin TRAP. Expansion microscopy revealed partial colocalization of CLAMP and TRAP in a subset of micronemes, and a distinct accumulation of CLAMP at the apical tip of sporozoites. Collectively, these results demonstrate that CLAMP is essential across invasive stages of the malaria parasite, and support a role of the protein upstream of host cell invasion, possibly by regulating the secretion or function of adhesins in *Plasmodium* sporozoites.

**Funding:** This work was funded by grants from the Laboratoire d'Excellence ParaFrap (ANR-11-LABX-0024 to OS), the Agence Nationale de la Recherche (ANR-20-CE18-0013 to OS) and the Fondation pour la Recherche Médicale (EQU201903007823 to OS). ML was supported by a 'DIM 1Health' doctoral fellowship awarded by the Conseil Régional d'Ile-de-France. The funders had no role in study design, data collection and analysis, decision to publish, or preparation of the manuscript.

**Competing interests:** The authors have declared that no competing interests exist.

## Author summary

*Plasmodium* parasites, the causative agents of malaria, are transmitted during the bite of an infected mosquito. Infectious parasite stages known as sporozoites are released from the insect salivary glands and injected into the host skin. Sporozoites rapidly migrate to the host liver, invade hepatocytes and differentiate into the next invasive forms, the merozoites, which invade and replicate inside red blood cells. Sporozoite motility and host cell invasion rely on the secretion of apical organelles called micronemes and rhoptries. Here we characterize the function of a microneme protein expressed both in merozoites and sporozoites, the claudin-like protein CLAMP. We used a conditional genome editing strategy in a rodent malaria model to generate CLAMP-deficient sporozoites. In the absence of CLAMP, sporozoites failed to invade mosquito salivary glands and mammalian hepatocytes, and showed defects in gliding motility and microneme secretion. Our data establish that CLAMP plays an essential role across *Plasmodium* invasive stages, and might represent a potential target for transmission-blocking antimalarial strategies.

## Introduction

The life cycle of *Plasmodium* parasites is complex and alternates between a vertebrate host and an *Anopheles* vector, in which parasites face several stages of differentiation and development. In mammals, infection begins when motile forms of the parasite known as sporozoites are injected in the skin during the bite of a blood-feeding infected mosquito. These sporozoites then migrate through the dermis until they reach a blood vessel and are then transported in the blood flow to the liver. There, sporozoites infect hepatocytes and differentiate into thousands of merozoites that are then released in the blood and invade red blood cells (RBCs). Sporozoites are formed in oocysts in the mosquito midgut, and once released in the haemolymph, they colonize the insect salivary glands in order to permit transmission upon salivation during bite.

Host cell invasion by *Plasmodium* and related apicomplexans relies on the sequential secretion of apical secretory vesicles, the micronemes and the rhoptries, and the formation of a structure called moving junction (MJ), which mediates parasite internalization and the formation of a replicative niche, the parasitophorous vacuole (PV). The MJ has been well characterised during host cell invasion by *Toxoplasma* tachyzoites and *Plasmodium* merozoites [1,2]. In contrast, the nature of the junction during sporozoite entry into hepatocytes remains elusive [3]. Known components of the MJ, such as AMA1 and RONs, are also expressed in sporozoites, and are required during invasion not only of hepatocytes but also of the mosquito salivary glands [4–6]. A genome-wide CRISPR screen performed in *T. gondii* has identified a novel putative component of the MJ, the Claudin-Like Apicomplexan Microneme Protein (CLAMP) [7]. CLAMP orthologs are present in all available apicomplexan genomes and no related sequence could be identified outside the phylum. However, topology predictions indicate structural similarities between CLAMP and the mammalian tight-junction proteins claudin-15 and claudin-19 [7]. CLAMP is found in *Toxoplasma* tachyzoite micronemes and at the MJ during invasion of host cells. Conditional knockdown of *clamp* gene affects *T. gondii* host cell invasion without disturbing egress or microneme secretion [7]. Similarly, depletion of CLAMP in *P. falciparum* leads to a drastic reduction of asexual blood stage growth [7]. CLAMP is also predicted to be essential in *P. berghei* based on genome-wide mutagenesis [8]. A more recent study focusing on the role of CLAMP in *Theileria equi* showed that CLAMP-specific antibodies inhibit invasion of equine erythrocytes [9].

While proteomic studies have documented that CLAMP is also expressed in sporozoites in multiple *Plasmodium* species [10–12], its role has never been assessed so far in these stages. One obstacle for genetic studies of factors like CLAMP is that genes that are essential in blood stages, the stages that are used for genetic manipulation, are refractory to conventional knock-out strategies. Nevertheless, several approaches for conditional gene deletion have emerged allowing the study of blood stage-essential genes in sporozoites. One of these approaches is based on the dimerisable Cre recombinase (DiCre), which can excise DNA sequences flanked by Lox sites in an inducible manner. The recombinase is split into two inactive subunits each fused to a rapamycin-ligand, which interact in the presence of rapamycin, restoring the Cre activity [13,14]. *Plasmodium* asexual blood stage parasites can be targeted prior transmission to mosquitoes, allowing deletion of the gene of interest and phenotypical analysis in subsequent stages of the parasite life cycle [15,16]. In this study, we used the DiCre system in *P. berghei* to study the role of CLAMP in sporozoites. Conditional deletion of *clamp* gene resulted in a dramatic decrease of sporozoite invasion of the mosquito salivary glands and of mammalian hepatocytes. This severe phenotype was associated with a major defect in sporozoite gliding motility and reduced shedding of TRAP. This study reveals that CLAMP is required for *Plasmodium* sporozoite motility and infectivity in both the mosquito and mammalian hosts, and illustrates the robustness of the DiCre system for conditional genome editing across the parasite life cycle.

## Results

### Conditional deletion of *clamp* gene in *P. berghei*

In order to study the role of CLAMP in *P. berghei* sporozoites, we first engineered a parasite line allowing conditional disruption of *clamp* gene using the DiCre system. We genetically modified a *P. berghei* line (PbDiCre) expressing the two DiCre components in addition to a mCherry fluorescence cassette [15] in order to generate a *clamp*cKO line where the CLAMP-coding sequence is fused to a 3xFlag tag and flanked by LoxN sites. For this purpose, we used a two-step strategy to introduce in two successive transfections LoxN sites upstream and downstream of *clamp* gene, respectively (**Figs 1A** and **S1**) [17]. A GFP cassette was introduced between *clamp* and the second LoxN site, to facilitate monitoring of gene excision by flow cytometry or fluorescence microscopy. Transfected parasites were selected with pyrimethamine and sorted by flow cytometry, and the final *clamp*cKO parasites were cloned by limiting dilution and injection into mice. The parasites were genotyped by PCR to confirm correct construct integration (**S1 Fig**). Transfected parasites were monitored by flow cytometry during the different selection steps in order to confirm the presence or absence of the GFP cassette (**Fig 1B**).

We then assessed the effects of rapamycin on *clamp*cKO parasites during blood-stage growth. Following intravenous injection of $10^6$ parasitized RBCs into mice, one cohort was treated with a single oral dose of rapamycin, while the other was left untreated. We then monitored the parasitaemia over time by flow cytometry. Rapamycin exposure led to a decrease of GFP fluorescence in *clamp*cKO parasites, confirming the efficiency of gene excision (**Fig 1B**), yet we fail to recover mCherry⁺/GFP⁻ parasite populations. Rapamycin-induced excision of *clamp* greatly reduced blood-stage growth in *clamp*cKO-infected mice, as compared to untreated parasites (**Fig 1C**), in agreement with an essential role for CLAMP in asexual blood stages [7,8].

### CLAMP is required for invasion of mosquito salivary glands

We next investigated the role of CLAMP in *P. berghei* mosquito stages. For this purpose, mosquitoes were fed on *clamp*cKO-infected mice, which were treated or not with rapamycin one

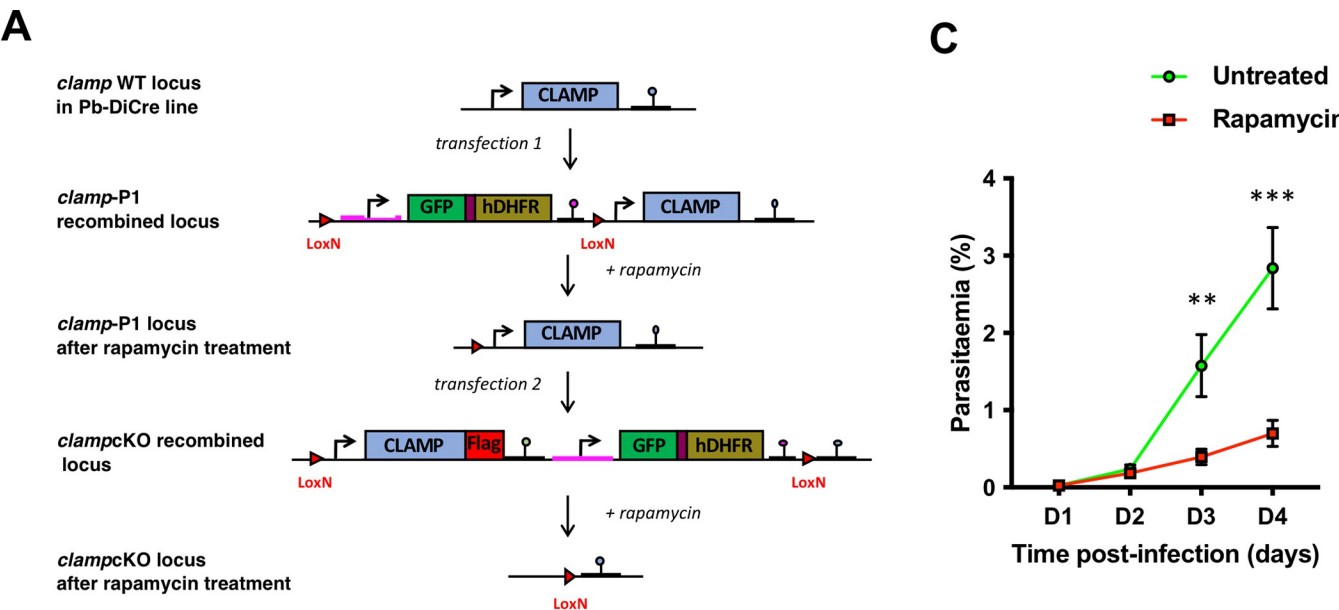

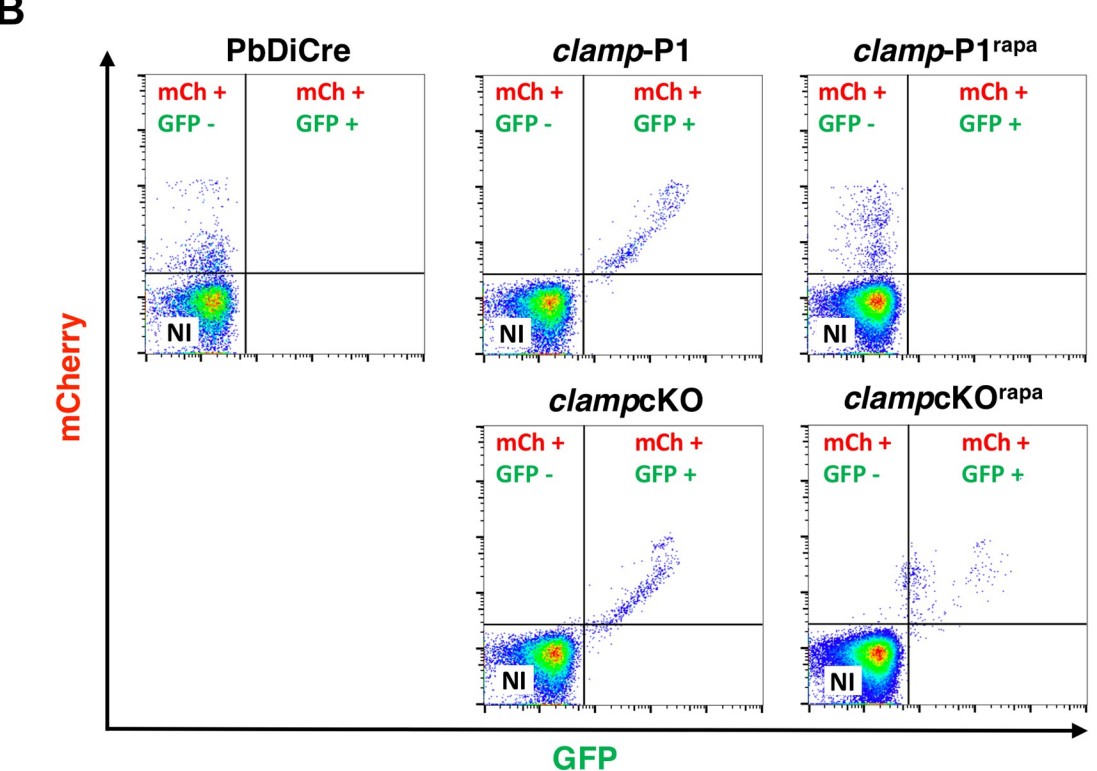

**Fig 1. Generation of *P. berghei clamp*cKO parasites. A.** Strategy to modify the *clamp* locus in the PbDiCre line. A first transfection in PbDiCre parasites with the P1 construct led to the insertion of a GFP-hDHFR expression cassette flanked by two LoxN site upstream of *clamp* gene. Rapamycin-induced excision of the cassette caused retention of one single LoxN site upstream of *clamp* in the treated *clamp*-P1 parasites. In a second step, rapamycin-treated excised *clamp*-P1 parasites were transfected with the P2 construct to generate a *clamp*cKO parasite line expressing CLAMP fused to a 3xFlag epitope tag in addition to GFP and hDHFR, and carrying two LoxN sites flanking *clamp* gene. Rapamycin-induced activation of DiCre leads to excision of the *clamp* gene together with the GFP-hDHFR cassette. **B.** Flow cytometry analysis of blood-stage parasites at different steps after initial transfection of the PbDiCre parental line to generate *clamp*cKO parasites. NI, non-infected red blood cells. **C.** Blood stage growth of rapamycin-treated and untreated *clamp*cKO parasites. Rapamycin was administered at day 1. The graph shows the parasitaemia (mean +/- SEM) in groups of 5 mice, as quantified by flow cytometry based on mCherry detection. **, p < 0.01; ****, p < 0.0001 (Two-way ANOVA). The experiment shown in panel C was only performed once.

day prior blood feeding, as described [15]. We then monitored parasite development in the mosquito midgut and colonization of the insect salivary glands using fluorescence microscopy. Analysis of dissected midguts at day 16 post-feeding showed that both untreated and rapamycin-exposed parasites produced mCherry-positive oocysts (**Fig 2A**). As expected, only untreated parasites retained the GFP fluorescence, while oocysts from rapamycin-treated parasites were GFP-negative, confirming efficient gene excision (**Fig 2A**). Mosquito salivary glands dissected at day 21 post-feeding exhibited a strong fluorescence with both mCherry and GFP, reflecting sporozoite accumulation in the glands (**Fig 2B**). Strikingly, salivary glands from rapamycin-exposed *clamp*cKO-infected mosquitoes showed a weak mCherry fluorescence signal, suggesting low parasite loads (**Fig 2B**). Sporozoites were then collected from midguts or salivary glands and counted. Rapamycin treatment of *clamp*cKO parasites had no major impact on the number of midgut sporozoites (**Fig 2C**), but severely reduced the number of salivary gland sporozoites (**Fig 2D**). As expected, rapamycin treatment induced robust gene excision in *clamp*cKO parasites, as determined based on the percentage of excised (mCherry$^+$/GFP$^-$) and non-excised (mCherry$^+$/GFP$^+$) sporozoites (**S2A** and **S2B Fig**). Reduced numbers of salivary gland sporozoites could result either from a defect in parasite egress from oocyst or from a defect in salivary gland invasion. We collected and counted haemolymph sporozoites in rapamycin-exposed and untreated *clamp*cKO-infected mosquitoes and observed no significant difference (**Fig 2E**). Mosquitoes infected with parasites expressing mCherry exhibit a red fluorescence of pericardial cells following uptake of sporozoites released in the haemolymph [4]. A large majority of infected mosquitoes presented such mCherry-labelled structures, with both rapamycin-exposed and untreated *clamp*cKO parasites (**S2C Fig**), indicating efficient egress from oocysts in CLAMP-deficient parasites. Together, these results show that sporozoites deleted for *clamp* are able to develop normally in the mosquito until release in the haemolymph but are then impaired in their ability to invade the salivary glands. The lack of accumulation of sporozoites in the haemolymph could be due to the slight reduction in midgut sporozoite numbers (**Fig 2C**) or, more likely, to the rapid elimination of sporozoites from the mosquito circulation [18].

## Rapamycin-induced gene excision abrogates CLAMP protein expression in sporozoites

In order to verify that *clamp* gene excision induced by rapamycin exposure was efficient in depleting CLAMP protein, we took advantage of the 3xFlag fused to the C-terminus of CLAMP in the *clamp*cKO line (**Fig 1A**). Analysis of untreated *clamp*cKO salivary gland sporozoites by immunofluorescence using anti-Flag antibodies revealed a punctate distribution of the protein in the cytoplasm of the parasite, consistent with a localization in the micronemes (**Fig 3A**). In contrast, no labelling was observed in rapamycin-exposed *clamp*cKO sporozoites, confirming complete depletion of the protein upon *clamp* gene excision (**Fig 3A**). We also confirmed by western blotting the absence of CLAMP protein in haemolymph sporozoites after rapamycin treatment, while in untreated parasites the protein was detected as a single ~40 kDa band (**Fig 3B**). These data thus confirm that CLAMP is expressed in *P. berghei* sporozoites and validate the conditional approach to deplete CLAMP protein in sporozoites.

## CLAMP is essential for sporozoite cell traversal and hepatocyte invasion

We then assessed the capacity of salivary gland sporozoites to traverse and invade hepatocytes *in vitro* in the absence of CLAMP. We first quantified by flow cytometry the number of traversed cells 3 hours after sporozoite addition to HepG2 cell cultures, using a dextran-based assay as previously described [19,20]. Cell traversal was severely impaired in sporozoites

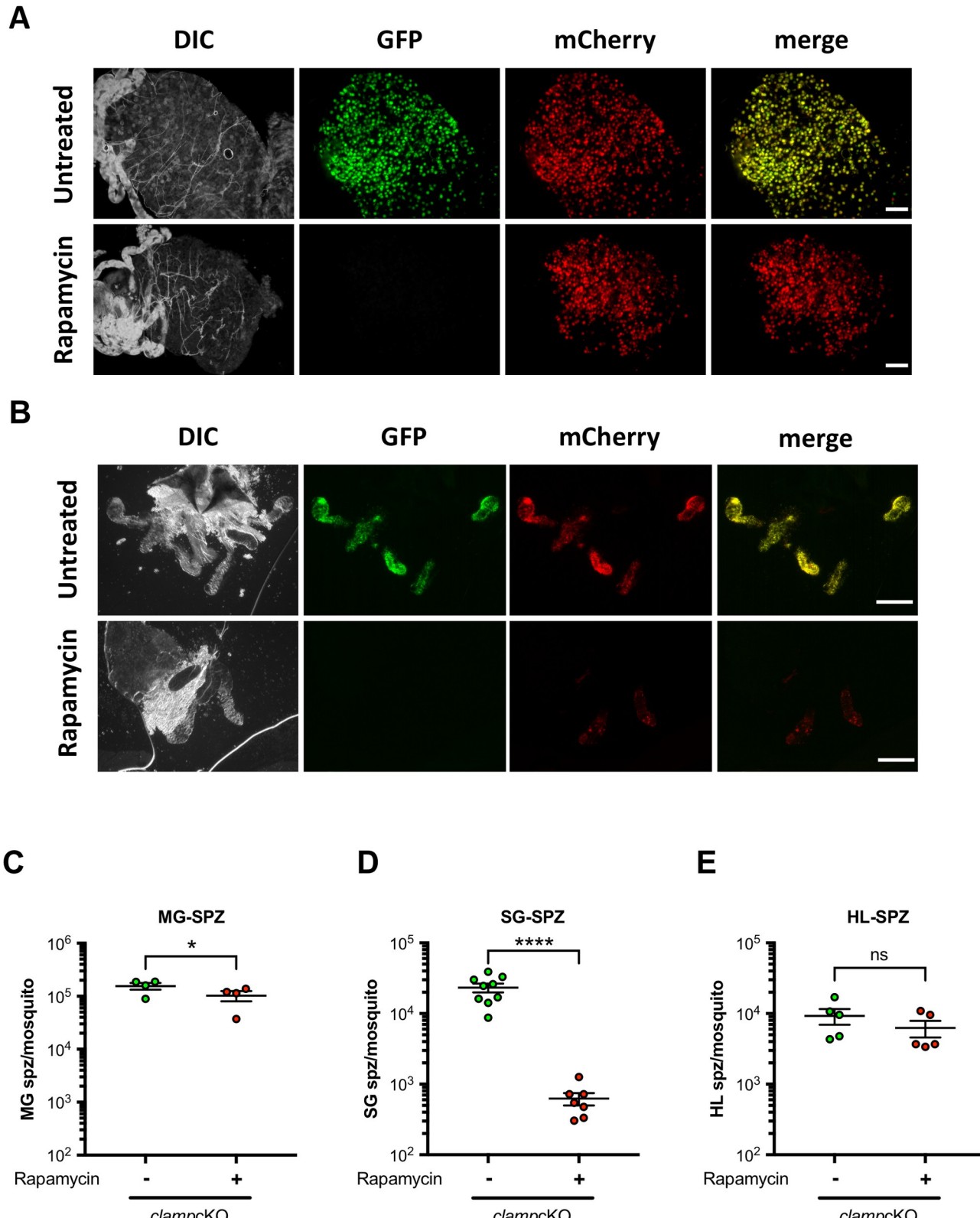

**Fig 2. Deletion of *clamp* impacts invasion of mosquito salivary glands by sporozoites. A-B.** Fluorescence microscopy imaging of midguts (A) and salivary glands (B) from rapamycin-exposed and untreated *clamp*cKO-infected mosquitoes, dissected at day 16 or 21 post-feeding, respectively. Exposure time and contrast were adjusted at the same level for each channel in both conditions. Scale bar, 200 μm. **C-E.** Comparison of sporozoite

numbers collected from midguts (C), salivary glands (D) and haemolymph (E) of female mosquitoes infected with rapamycin-exposed and untreated *clamp*cKO parasites. All results shown in C-E are mean +/- SEM of at least four independent experiments. Ns, non-significant; *, p < 0.05; ****, p<0.0001 (Two-tailed ratio paired t test).

lacking CLAMP, as shown by a dramatic reduction of the percentage of dextran-positive cells in cultures incubated with rapamycin-exposed *clamp*cKO as compared to untreated *clamp*cKO sporozoites (**Fig 4A**). We next tested if sporozoites lacking CLAMP could infect and develop into EEFs in cell cultures. HepG2 cells were incubated with rapamycin-exposed and untreated *clamp*cKO sporozoites and EEFs were quantified at 24h post-infection by microscopy after staining of PVs with antibodies against UIS4, a marker of the PV membrane [21]. These experiments revealed an almost complete absence of EEFs in cultures inoculated with rapamycin-exposed *clamp*cKO parasites (**Fig 4B**). These data show that, in addition to its role during salivary gland invasion in the mosquito, CLAMP is also essential in sporozoites for cell traversal and productive invasion of mammalian cells.

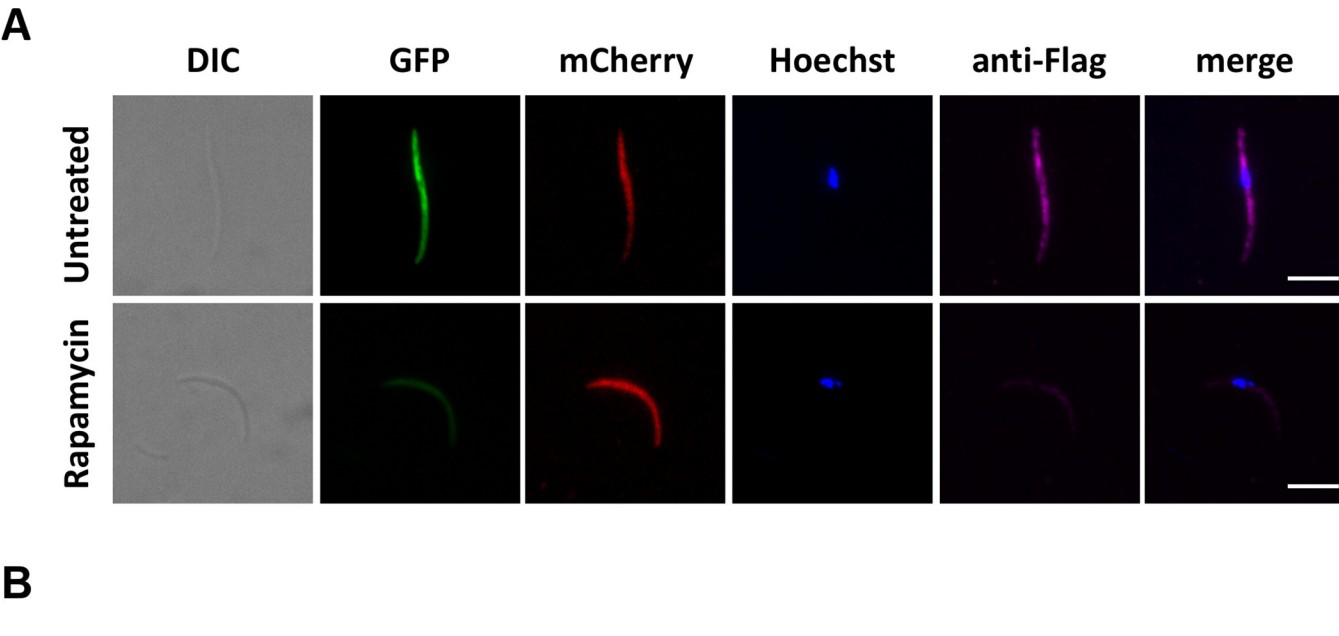

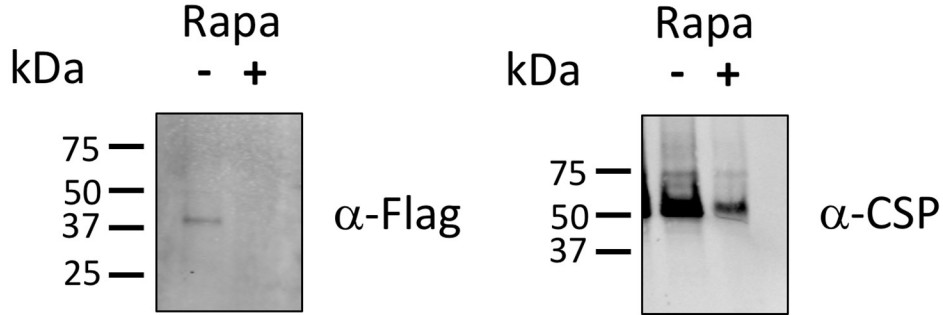

**Fig 3. Conditional gene deletion abrogates CLAMP protein expression in sporozoites. A.** Immunofluorescence analysis of rapamycin-exposed and untreated *clamp*cKO sporozoites labelled with anti-Flag antibodies (magenta). Both rapamycin-exposed and untreated *clamp*cKO parasites express mCherry (red), while only untreated *clamp*cKO parasites express GFP (green). Nuclei were stained with Hoechst 77742 (blue). Scale bar, 5 μm. **B.** Western bot analysis of haemolymph sporozoite lysates from untreated or rapamycin-treated parasites, using anti-Flag antibodies to detect CLAMP. CSP was used as a loading control.

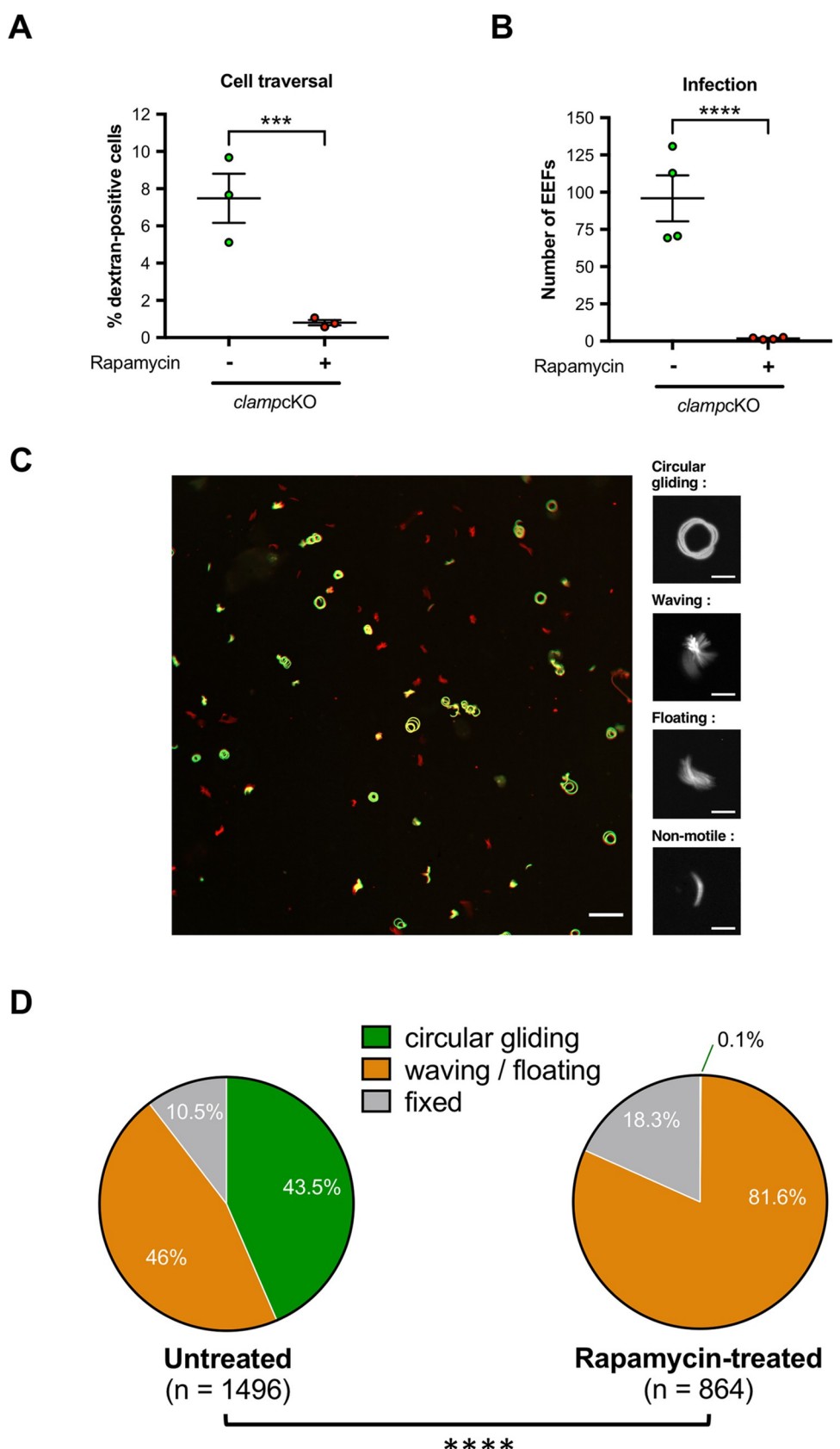

**Fig 4. CLAMP-deficient sporozoites are impaired in transcellular migration, host cell infection and motility. A.** Quantification of traversed (dextran-positive) HepG2 cells by FACS after incubation for 3h with rapamycin-exposed and untreated *clamp*cKO salivary gland sporozoites in the presence of fluorescein-labelled dextran. Results shown are mean +/- SEM of three independent experiments, each performed with 5 technical replicates. **B.** Quantification of UIS4-labelled exo-erythrocytic forms (EEFs) in HepG2 cells as determined by fluorescence microscopy 24h post-invasion with rapamycin-exposed and untreated *clamp*cKO salivary gland sporozoites. Results shown are mean +/- SEM of four independent experiments with 5 technical replicates for each. ***, $p<0.001$; ****, $p<0.0001$ (Two-tailed ratio paired t test). **C.** Maximum intensity projection of video-microscopy images of untreated *clamp*cKO (mCherry[+]/GFP[+], yellow) and rapamycin-exposed *clamp*cKO (mCherry[+]/GFP[-], red) salivary gland sporozoites, recorded for 3 min. Sporozoites were mixed and activated at 37°C in the presence of albumin before imaging. Their movement patterns were classified in four categories: circular gliding, waving, floating and fixed (shown in magnified images). Scale bars, 50 μm for projection and 10 μm for magnified images. **D.** Quantification of motility patterns in untreated (n = 1496) and rapamycin-exposed (n = 864) *clamp*cKO sporozoites. Sporozoites were classified in three groups based on their motility pattern. ****, $p<0.0001$ (Chi-square).

## CLAMP is essential for sporozoite gliding motility

Since cell traversal and productive host cell invasion both rely on the parasite gliding machinery, we hypothesized that the combined phenotype observed with parasites lacking CLAMP could be caused by a defect in sporozoite motility. To address this hypothesis, we analyzed the motility of rapamycin-exposed and untreated *clamp*cKO by video-microscopy. One limitation of the assessment of sporozoite motility was the medium of dissection that was enriched in mosquito debris in the case of rapamycin-exposed parasites as compared to untreated parasites, due to the highly reduced number of salivary gland sporozoites. To ensure that the differences in the composition of the medium would not affect the motility of the parasite, we mixed untreated *clamp*cKO (mCherry[+]/GFP[+]) and rapamycin-exposed *clamp*cKO (mCherry[+]/GFP[-]) sporozoites and imaged the mixed population (**S1 Movie**). Sporozoite motility was classified in three groups based on the movement patterns: circular gliding, waving or floating, or fixed (non-motile) (**Fig 4C**). The effect of *clamp* gene deletion was striking, as over 864 observed *clamp*cKO sporozoites in the rapamycin-treated condition, only 1 (0,1%) was motile and showed a circular gliding pattern. The vast majority of rapamycin-exposed *clamp*cKO were observed waving or floating (81,6%) (**Fig 4D** and **S1 Movie**). In comparison, more than 40% of untreated *clamp*cKO sporozoites were motile and exhibited circular gliding (43,5%) (**Fig 4D** and **S1 Movie**). These results show that circular gliding is abrogated in sporozoites in the absence of CLAMP. This phenotype could explain the defects observed in salivary gland invasion and in cell traversal and hepatocyte invasion, as motility is needed by the parasite to actively enter or exit cells.

## CLAMP is required for TRAP shedding

In order to get more insights into the function of CLAMP, we searched for potential interacting partners of the protein in sporozoites through immunoprecipitation (IP) followed by mass spectrometry (MS). IP was performed using anti-Flag antibodies on two independent lysates from untreated *clamp*cKO salivary gland sporozoites. Control IP was performed using lysates from untagged PbGFP sporozoites. Analysis of samples by MS revealed 7 proteins that were identified in the two co-IP experiments but not in control IP experiments: CLAMP, the thrombospondin-related anonymous protein (TRAP, PBANKA_1349800), a putative pantothenate transporter (PAT, PBANKA_0303900), the elongation factor 1-alpha (PBANKA_1133300), the tubulin beta chain (PBANKA_1206900), actin I (PBANKA_1459300), and a conserved protein of unknown function (PBANKA_0403700) (**Fig 5A** and **S1 Table**). The co-purification of TRAP with CLAMP was confirmed by western blotting in an independent co-IP experiment (**Fig 5B**). Elongation factor 1-alpha is likely a contaminant in the pulldown experiments,

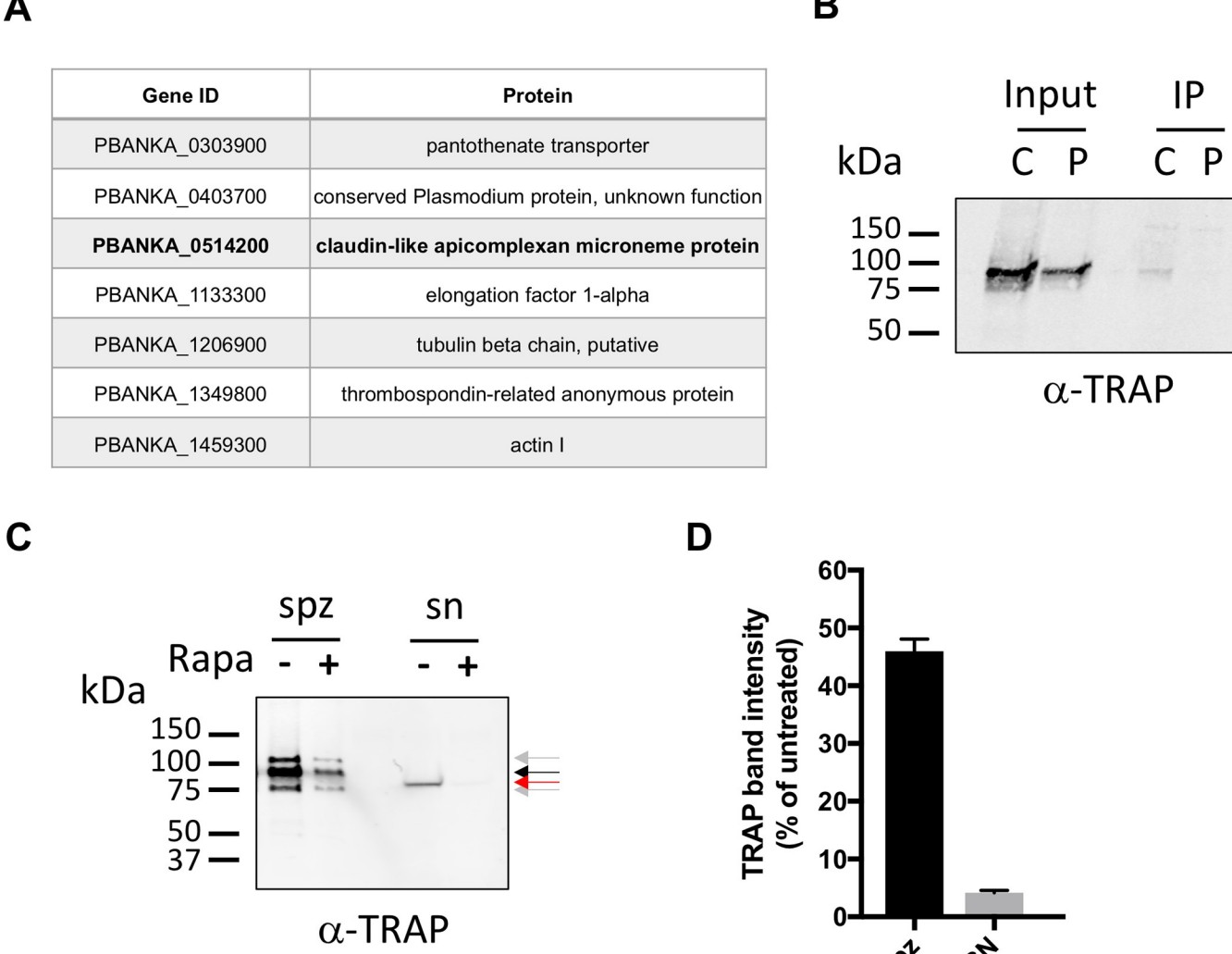

**Fig 5. CLAMP is essential for TRAP shedding. A.** *P. berghei* proteins identified by mass spectrometry after anti-Flag immunoprecipitation from two independent CLAMP-flag sporozoite lysates, and not from control samples. **B.** Additional lysates were prepared from CLAMP-flag and parental *P. berghei* salivary gland sporozoites, and used for anti-Flag IP followed by western blotting using anti-TRAP antibodies. TRAP was detected in total extracts (input) from both CLAMP-flag (C) and parental (P) samples, but only in CLAMP-Flag immunoprecipitate (IP). **C.** TRAP secretion assay using haemolymph sporozoites (5 x $10^4$ or equivalent per lane) from untreated or rapamycin-treated *clamp*cKO parasites. Microneme secretion was stimulated by incubation for 15 min at 37°C in the presence of BSA and ethanol. After stimulation of microneme secretion, samples were fractionated by centrifugation in sporozoite pellets (spz) and supernatants containing secreted proteins (SN), and analyzed by western blot using anti-TRAP antibodies. TRAP protein was detected as a ~100 kDa major band (black arrow) and two minor bands (grey arrows) in sporozoite lysates, and as a single ~75 kDa in supernatants (red arrow). **D.** Quantification of signal intensity of TRAP western blot bands in sporozoite pellets (main band only) and supernatants. The data show TRAP levels in rapamycin-treated parasites relative to untreated parasites, both in pellets and supernatants.

which might also be the case for tubulin. In contrast, the identification of TRAP, PAT and actin in CLAMP immunoprecipitates provides a potential link to CLAMP function.

TRAP is a sporozoite adhesin secreted from micronemes that connects extracellular surfaces to the parasite intracellular actin-myosin motor machinery and is essential for sporozoite gliding motility [22,23]. Interestingly, PAT has been shown to regulate exocytosis of osmiophilic bodies and micronemes in *P. berghei* gametocytes and sporozoites, respectively [24]. Sporozoites lacking PAT fail to secrete TRAP, and, as a result, are immotile and thus unable to infect host cells [24]. Since the phenotype of CLAMP-deficient sporozoites is reminiscent of that of

TRAP- and PAT-deficient parasites, and based on our co-IP data suggesting a potential interaction between these proteins, we investigated whether CLAMP could be involved in TRAP secretion. For this purpose we used haemolymph sporozoites as the number of CLAMP-deficient salivary gland sporozoites was too low to allow western blot experiments. Rapamycin-exposed and untreated *clamp*cKO haemolymph sporozoites were incubated at 37°C in the presence of albumin and ethanol to stimulate microneme secretion [25], and shedding of TRAP in the supernatant (SN) was assessed by western-blot. TRAP was detected in sporozoite lysates (pellet) from both rapamycin-exposed and untreated *clamp*cKO parasites as a main band around 100kDa, which corresponds to the expected size of the mature protein in *P. berghei* [22,26]. Two additional minor bands were also detected, likely reflecting different forms of the protein in haemolymph sporozoites (**Fig 5B**). A shed form of TRAP was detected in the supernatant of untreated *clamp*cKO sporozoites as a band around 75kDa, but was nearly absent in the supernatant of rapamycin-exposed *clamp*cKO sporozoites, revealing a defect of TRAP shedding in parasites lacking CLAMP (**Fig 5B**). Quantification of band intensity showed a ~90% reduction of TRAP release in the supernatant as compared to sporozoite-associated protein (**Fig 5C**). These results strongly suggest that CLAMP drives sporozoite motility by regulating the secretion and/or cleavage of TRAP.

## CLAMP is localized in a subset of micronemes and accumulates at the apical tip of sporozoites

Finally, we analyzed in more details the localization of CLAMP in sporozoites, in relation with TRAP. For this purpose, we employed ultrastructure expansion microscopy (U-ExM) [27], which has recently emerged as a powerful technique to image subcellular compartments at high resolution in Apicomplexa [28–30]. Sporozoites were collected from the salivary glands of mosquitoes infected with CLAMP-flag (untreated *clamp*cKO) parasites, expanded in gels and imaged following labelling with antibodies specific for Flag or the micronemal proteins TRAP and AMA1. The shape and structure of sporozoites were well preserved, with an expansion rate of about 4x (**Figs 6** and **S3**). Flag-tagged CLAMP localized in numerous vesicles throughout the cytoplasm of the parasite, showing a typical micronemal pattern as also observed with TRAP (**Figs 6A** and **S3A**). CLAMP and TRAP colocalized in a subset of micronemes, as evidenced by labelling by both anti-Flag and anti-TRAP antibodies (**Fig 6A**). In contrast, the distribution of CLAMP and AMA1 showed limited overlap (**Figs 6B** and **S3B**), suggesting that these proteins localize to distinct microneme populations. However, the resolution was not sufficient to allow reliable quantification of microneme subsets, given the high density of vesicles in some portions of the sporozoites. Most interestingly, U-ExM revealed a clear accumulation of CLAMP at the apical tip of salivary gland sporozoites (**Figs 6** and **S3**). This apical focal staining was observed in a vast majority of the parasites (23 out of 28 analyzed, 82%), and was absent after rapamycin-induced deletion of *clamp* gene, ruling out a staining artefact (**S3C Fig**). These observations are consistent with our functional data showing that CLAMP regulates TRAP secretion and/or function in sporozoites, and raise the possibility that CLAMP plays additional roles at the site of organelle secretion at the apex of the parasite.

## Discussion

We report here the first functional characterization of CLAMP in *Plasmodium* sporozoites, revealing its crucial role in gliding motility and infectivity in both the mosquito and mammalian hosts. We generated a PbDiCre line in which *clamp* gene and a GFP expression cassette were flanked by LoxN sites. We first validated the efficient excision of *clamp* gene in blood stages, based on the disappearance of the GFP fluorescence upon exposure to rapamycin, yet

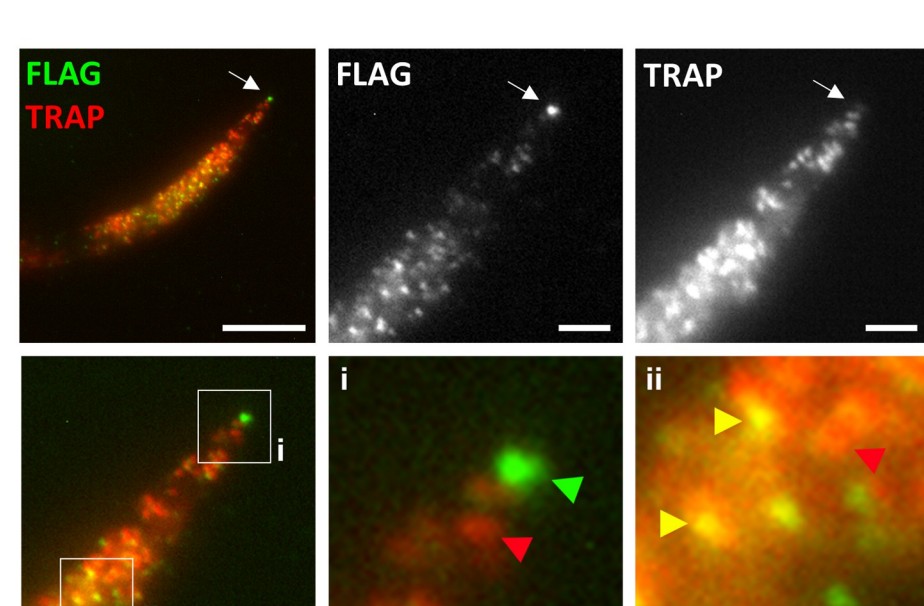

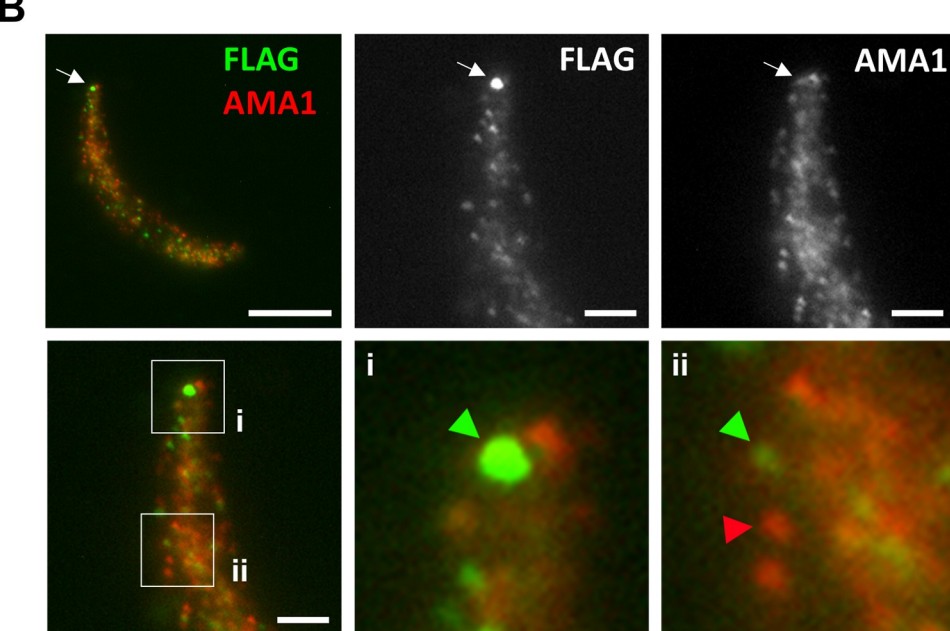

**Fig 6. Expansion microscopy reveals a unique distribution of CLAMP in sporozoites. A.** Salivary gland sporozoites expressing Flag-tagged CLAMP (untreated *clamp*cKO parasites) were examined by expansion microscopy using antibodies against Flag (green) and TRAP (red). Specific accumulation of CLAMP is clearly visible at the apical tip of sporozoites, as indicated by arrows in the upper panels. The lower i and ii magnified panels illustrate the distinct distribution of CLAMP (green arrowheads) and TRAP (red arrowheads), and the colocalization of both proteins in a subset of micronemes (yellow arrowheads). Scale bars, 10 μm (upper left panel) or 2.5 μm (other panels). **B.** Sporozoites labeled with antibodies against Flag (green) and AMA1 (red) were examined by expansion microscopy. Apical accumulation of CLAMP is indicated by arrows in the upper panels. The lower i and ii magnified panels illustrate the distinct distribution of CLAMP (green arrowheads) and AMA1 (red arrowhead). Scale bars, 10 μm (upper left panel) or 2.5 μm (other panels).

we failed to establish stable CLAMP-deficient (GFP-negative) parasite populations, consistent with CLAMP being essential in blood stages [7,8]. Transmission of CLAMP-deficient parasites to mosquitoes revealed that in the absence of the protein sporozoites develop normally and egress from oocysts, but fail to invade the salivary glands. Nevertheless, despite the low numbers of salivary gland sporozoites, we could analyze their phenotype in functional assays, which revealed that, in addition to the defect in salivary gland invasion in the mosquito, CLAMP-deficient sporozoites are impaired in both traversal and invasion of mammalian hepatocytic cells, associated with disruption of gliding motility.

The severe defect in gliding motility could on its own explain the phenotype of CLAMP-deficient sporozoites, since motility is required for both cellular transmigration and host cell invasion. Sporozoite proteins specifically involved in cell traversal, such as SPECT and PLP1, differ from CLAMP as they are not required for motility, invasion of the mosquito salivary glands or productive invasion of liver cells, at least *in vitro* [31–33]. Reciprocally, proteins acting specifically during host cell invasion, such as the 6-cysteine domain proteins P36 and P52, are not required for gliding motility and cell traversal [34–36]. CLAMP-deficient sporozoites also have a distinct phenotype as compared to parasites lacking AMA1, a canonical component of the MJ. Although in both cases sporozoites show a defect in invasion of mosquito salivary glands and mammalian hepatocytes, AMA1 conditional mutant sporozoites have no defect in gliding motility and cell traversal [4]. Altogether, these observations strongly support a role of CLAMP upstream of host cell invasion and MJ formation in *Plasmodium* sporozoites, unlike previously observed in *Toxoplasma* tachyzoites [7].

Like in other apicomplexan invasive stages, sporozoite motility relies on a multimolecular machinery called the glideosome, which is localized underneath the parasite pellicle and links the parasite actin-myosin motor to surface adhesins that interact with extracellular substrates [37]. The micronemal protein TRAP is an adhesin that plays a pivotal role in sporozoite gliding motility. Disruption of *trap* gene abrogates sporozoite motility and prevents invasion of the mosquito salivary glands as well as mammalian hepatocytes [23,38,39]. Several lines of evidence support the existence of a functional link between CLAMP and TRAP-mediated gliding motility. TRAP was identified as a potential CLAMP-interacting partner in co-IP experiments, and colocalizes with CLAMP in a subset of micronemes in salivary gland sporozoites. Additionally, TRAP shedding is impaired in CLAMP-deficient haemolymph sporozoites. Whether CLAMP participates in a multimolecular complex with TRAP and possibly other sporozoite components remains to be established. Future studies will also determine how precisely CLAMP regulates TRAP shedding. CLAMP could associate with TRAP to regulate its trafficking and/or secretion, or be involved in signalling pathways leading to microneme secretion. Alternatively, it might regulate the proteolytic processing of TRAP at the sporozoite surface following microneme exocytosis [22]. Whether CLAMP controls the secretion of other microneme proteins in *Plasmodium* sporozoites also deserves further investigation. In this regard, our U-ExM data showing partial co-distribution of CLAMP and TRAP, but not AMA1, support the existence of distinct microneme populations in *Plasmodium* sporozoites, with possibly distinct trafficking and/or signalling pathways controlling microneme exocytosis [40,41].

Co-IP and mass spectrometry also identified the pantothenate transporter PAT as a putative CLAMP-interacting partner in sporozoites. PAT has been shown to be required for TRAP secretion and sporozoite motility in *P. berghei*, and plays an additional role in gametocytes, where it regulates exocytosis of osmiophilic bodies during egress [24]. Therefore, PAT is essential for parasite transmission to the mosquito [24,42,43]. In our experimental setup, DiCre-mediated excision of *clamp* gene can be induced only shortly before transmission, preventing the analysis of CLAMP role in gametocytes or ookinetes. In *Toxoplasma* tachyzoites, the homolog of PAT, called Transporter Facilitator Protein 1 (TFP1), is required for microneme

maturation [44]. Conditional depletion of TFP1 impairs microneme biogenesis and leads to a complete block in exocytosis. Whether PAT plays a similar role in *Plasmodium* is unknown.

In addition to TRAP, various proteins have been identified as involved in sporozoite gliding motility, such as the TRAP-related protein (TREP) [45,46], LIMP [47] or the Secreted Protein with Altered Thrombospondin Repeat (SPATR) [48]. Like CLAMP, SPATR is conserved in Apicomplexa and is essential in asexual blood stages of the parasite life cycle. Interestingly, a recent study based on conditional knockdown in *P. berghei* showed that sporozoites lacking SPATR have impaired motility, strongly reduced capacity to invade salivary glands and decreased infectivity to mice [48], a phenotype that is similar to the one we observed with CLAMP-deficient parasites. Costa *et al.* excluded a role of SPATR in TRAP secretion, based on the detection of TRAP at the surface of mutant sporozoites by immunofluorescence [48]. In our hands, surface staining of TRAP did not show any overt difference between untreated and rapamycin-treated parasites, irrespective of stimulation of microneme secretion (**S4 Fig**). It is possible however that surface immunostaining may not capture the entire dynamics of TRAP secretion and shedding.

In *Toxoplasma* tachyzoites, depletion of CLAMP does not impact the secretion and shedding of MIC2, a micronemal adhesin related to TRAP [7], suggesting that the protein may have different functions in *Toxoplasma* and *Plasmodium* parasites. CLAMP forms a complex with two other micronemal proteins in *Toxoplasma* tachyzoites, SPATR and CLIP (CLAMP-interacting protein) [49]. Depletion of CLAMP specifically impairs the secretion of SPATR and CLIP, but not of other micronemal proteins. Importantly, both CLAMP and CLIP are required in tachyzoites for rhoptry discharge and host cell invasion [49]. Interestingly, SPATR is essential in *Plasmodium* but dispensable in *Toxoplasma* [48–50], and, like CLAMP, is required for gliding motility in *Plasmodium* sporozoites but not *Toxoplasma* tachyzoites [7,48–50]. The phenotypical differences observed with CLAMP and SPATR mutants in *Plasmodium* sporozoites versus *Toxoplasma* tachyzoites could reflect species-specific functional diversification or stage-specific functions of CLAMP and SPATR, considering the highly migratory behavior of *Plasmodium* sporozoites. Of note, CLIP is also expressed in *Plasmodium* sporozoites [10,11], but its role in this stage remains unknown.

Remarkably, U-ExM revealed a unique accumulation of CLAMP at the apical tip of sporozoites. While we cannot formally exclude a redistribution of the protein during sample preparation for U-ExM, this localization is reminiscent of the recently described Nd6 and Cysteine Repeat Modular Proteins (CRMPs) in *Toxoplasma*, which localize to secretory vesicles throughout the cytoplasm and additionally accumulate at the apex of extracellular tachyzoites [51,52]. Like CLAMP, Nd6 and CRMPs are required for rhoptry discharge and host cell invasion in *Toxoplasma*, but their molecular function is currently unknown [51–53]. Interestingly, immuno-EM documented the localization of Nd6 to an apical vesicle at the site of rhoptry discharge [51]. The focal accumulation of CLAMP at the apex of *Plasmodium* sporozoites raises the possibility that CLAMP has additional roles in sporozoites besides the regulation of TRAP-mediated gliding motility, possibly in rhoptry discharge and host cell invasion. Because the gliding machinery is also required for host cell invasion, it is difficult to disentangle the two processes with currently available tools. Alternatively, CLAMP could be involved in common signalling pathways involved in both microneme and rhoptry secretion in sporozoites. In this regard, CLAMP harbours a C-terminal proline-rich domain that is exposed to the cytosol [49], where it could interact with components of trafficking and/or signalling pathways.

In summary, we show here that CLAMP plays an essential role across invasive stages of the malaria parasite, and is required for gliding motility and infectivity of *P. berghei* sporozoites, notably by regulating the secretion of TRAP. CLAMP may thus represent a potential new target for antimalarial strategies. In this regard, a recent study showed that CLAMP-specific

antibodies can inhibit invasion of equine erythrocytes by the apicomplexan *Theileria equi*, and identified neutralization-sensitive epitopes in the predicted extracellular loops of the protein [9]. It will be important to test whether *Plasmodium* CLAMP can be similarly targeted by neutralizing antibodies. This would open interesting perspectives for interventions acting both against merozoites, to prevent invasion of erythrocytes, and sporozoites, to block infection early after transmission by the mosquito.

## Materials and methods

### Ethics statement

All animal work was conducted in strict accordance with the Directive 2010/63/EU of the European Parliament and Council on the protection of animals used for scientific purposes. Protocols were approved by the Ethical Committee Charles Darwin N˚005 (approval #7475–2016110315516522).

### Experimental animals, parasites and cell lines

Female Swiss mice (6–8 weeks old, from Janvier Labs) were used for all routine parasite infections. Parasite transfections were performed in the parental PbDiCre line (ANKA strain), which constitutively express the DiCre components in addition to a mCherry fluorescence cassette [15]. Parasite infections in mice were initiated through intraperitoneal injections of infected RBCs. A drop of blood was collected from the tail in 1ml PBS daily and used to monitor the parasitaemia by flow cytometry. *Anopheles stephensi* mosquitoes were reared at 24˚C with 80% humidity and permitted to feed on anaesthetised infected mice, using standard methods of mosquito infection as previously described [54]. Post-feeding, *P. berghei*-infected mosquitoes were kept at 21˚C and fed on a 10% sucrose solution. Midgut and haemolymph sporozoites were collected at day 16 and salivary gland sporozoites at day 21 post-infection. Sporozoites were collected by hand dissection and homogenisation of isolated midguts or salivary glands in complete DMEM medium (supplemented with 10% FBS, 1% Penicillin-Streptomycin and 1% L-Glutamine), then counted in a Neubauer haemocytometer. HepG2 cells (ATCC HB-8065) were cultured in complete DMEM in collagen-coated plates, at 37˚C, 5% $CO_2$, as previously described [55], and used for invasion and cell traversal assays.

### Generation of the *clamp*cKO parasite line in *P. berghei*

*Plasmid constructs*. Two plasmids, P1 and P2, were generated for insertion of LoxN sites upstream and downstream of *clamp*, respectively. The P1 plasmid was assembled by inserting two homologous sequences, 5'HR1 and 5'HR2, both localized upstream of *clamp* promoter region, in the pUpstream2Lox plasmid (Addgene #164573), which contains a GFP-2A-hDHFR cassette flanked by two LoxN sites. Both 5'HR1 (895 bp) and 5'HR2 (1014 bp) were amplified by PCR from WT PbANKA genomic DNA, and inserted into *Kpn*I/*Xho*I and *Nhe*I sites, respectively, of the pUpstream2Lox plasmid. The P2 plasmid was assembled by inserting two homologous sequences, 3'HR1 and 3'HR2, corresponding to the end of *clamp* ORF and to *clamp* 3'UTR, respectively, in the pDownstream1Lox plasmid (Addgene #164574), which contains a GFP-2A-hDHFR cassette followed by a single LoxN site. Both 3'HR1 (862 bp) and 3'HR2 (999 bp) were amplified by PCR from WT PbANKA genomic DNA, and inserted into *Kpn*I/*Xho*I and *Nhe*I sites, respectively, of the pDownstream1Lox plasmid. A triple Flag epitope tag was inserted in frame with *clamp* ORF immediately before the stop codon. In addition, a 559 bp fragment corresponding to the 3' UTR sequence from *P. yoelii clamp* gene (PY17X_0515300) was inserted immediately downstream of 3'HR1, to allow proper gene

expression and avoid spontaneous recombination with the 3' UTR of *P. berghei* clamp, which was used as 3'HR2. Plasmids P1 and P2 were verified by Sanger DNA sequencing (Eurofins Genomics) and linearized with *Kpn*I and *Nhe*I before transfection. All the primers used for plasmid assembly are listed in **S2 Table**.

*Transfection and selection*. Parental DiCre parasites were transfected with the P1 plasmid to generate *clamp*-P1 parasites, which were then exposed to rapamycin and transfected with the P2 plasmid to generate the final *clamp*cKO line. For the first transfection, schizonts purified from an overnight culture of PbDiCre blood stage parasites were transfected with 10 μg of linearized P1 plasmid using the AMAXA Nucleofector device (program U033), as previously described [56], and immediately injected intravenously into the tail vein of SWISS mice. To permit the selection of resistant transgenic parasites, pyrimethamine (35 mg/L) and 5-flurocytosine (0.5 mg/ml) were added to the mouse drinking water, starting one day after transfection. The parasitaemia was monitored daily by flow cytometry and the mice sacrificed at a parasitaemia of 2–3%, allowing preparation of frozen parasite stocks and isolation of parasites for genomic DNA extraction. After drug selection, GFP+/mCherry+ *clamp*-P1 parasites were sorted by flow cytometry. Mice were injected intraperitoneally with frozen parasite stocks and monitored until the parasitaemia was between 0.1 and 1%. On the day of sorting, one drop of tail blood was collected in 1ml PBS and used for sorting of 100 iRBCs on a FACSAria II (Becton-Dickinson), as described [57]. Sorted parasites were recovered in 200 μl RPMI containing 20% FBS and injected intravenously into two mice. *clamp*-P1 parasites were exposed to a single dose of rapamycin to induce excision of the GFP-2A-hDHFR cassette. The resulting GFP⁻/mCherry+ *clamp*-P1 parasites, which retained a single LoxN site inserted upstream of *clamp*, were sorted by flow cytometry and amplified in mice. Rapamycin-exposed *clamp*-P1 parasites were then transfected with the P2 plasmid, as described above, to generate the *clamp*cKO line. GFP+/mCherry+ *clamp*cKO parasites were sorted by flow cytometry, as described above, and cloned by limiting dilution and injection into mice to generate the final population used in this study. Parasites were genotyped at each step to verify the correct integration of the constructs and the absence of non-recombined *clamp* locus.

## Genotyping PCR

The blood collected from infected mice was passed through a CF11 column (Whatman) to deplete leucocytes. The collected RBCs were then centrifuged and lysed with 0.2% saponin (Sigma), before genomic DNA isolation using the DNA Easy Blood and Tissue Kit (Qiagen), according to the manufacturer's instructions. Specific PCR primers were designed to check for wild-type and recombined loci and are listed in **S2 Table**. PCR reactions were carried out using Recombinant Taq DNA Polymerase (Thermo Scientific) and standard PCR cycling conditions.

## Rapamycin-induced gene excision

*clamp*cKO-infected mice were administered a single dose of 200 μg rapamycin (Rapamune, Pfizer) by oral gavage. Treatment efficacy was validated by the observation of a decrease in GFP fluorescence intensity in circulating blood-stage parasites after treatment, reflecting gene excision. Rapamycin-treated *clamp*cKO parasites were transmitted to mosquitoes one day after rapamycin administration to mice, as described [15,17].

### *In vitro* infection assays

HepG2 cells cultured in DMEM complete medium were seeded at a density of 30,000 cells/well in a 96-well plate for flow cytometry analysis or 100,000 cells/well in 96-well μ-slide

(Ibidi) for immunofluorescence assays, 24 hours prior to addition of sporozoites. Culture medium was refreshed with complete DMEM on the day of infection, followed by incubation with 3,000 or 1,000 sporozoites, for flow cytometry or immunofluorescence assays, respectively. For quantification of traversal events, fluorescein-conjugated dextran (0.5 mg/ml, Life Technologies) was added to the wells together with sporozoites. After 3 hours, cells were washed twice with PBS, trypsinized, then resuspended in complete DMEM for analysis by flow cytometry on a Guava EasyCyte 6/2L bench cytometer equipped with 488nm and 532nm lasers (Millipore). Control wells were prepared without sporozoites to measure the basal level of dextran uptake. To quantify parasite liver stage infection, cells were washed twice with complete DMEM 3 hours after sporozoite addition, and then incubated for another 24h. Cultures were then fixed with 4% PFA, followed by two washes with PBS. Cells were then quenched with 0.1M glycine for 5 min, washed twice with PBS then permeabilized with 1% Triton X-100 for 5 min before 2 washes in PBS and blocking in PBS + 3% BSA. Samples were then stained with goat anti-UIS4 primary antibodies (1:500, Sicgen), followed by donkey anti-goat Alexa Fluor 594 secondary antibodies (1,1000, Life Technologies), both diluted in PBS with 3% BSA. EEFs were then counted based on the presence of a UIS4-stained PV.

## Imaging of parasites

Midguts and salivary glands were carefully collected from infected mosquitoes, mounted in PBS and directly examined without fixation on a fluorescence microscope. Sporozoites were collected by disruption of midguts or salivary glands and resuspended in PBS, placed on a coverslip then fixed for 10 min with 4% PFA followed by two washes with PBS. For immunostaining, sporozoites were fixed and permeabilized, and incubated with 3% BSA in PBS for 1h. Sporozoites were then stained with anti-Flag primary mouse antibody (M2 clone, Sigma), washed twice with PBS then incubated with Alexa Fluor anti-mouse 647 secondary antibodies (Life Technologies) and Hoechst 77742 (Life Technologies), before examination by fluorescence microscopy. All images were acquired on an Axio Observer Z1 Zeiss fluorescence microscope using the Zen software (Zeiss). The same exposure time was set for rapamycin-exposed and untreated *clamp*cKO parasites in order to allow comparisons. Images were processed with ImageJ for adjustment of contrast.

## Motility assay

For motility assays, rapamycin-exposed and untreated *clamp*cKO sporozoites were collected from manual dissection of infected mosquitoes and mixed at a 1:1 ratio, and kept in PBS on ice until imaging in PBS with 3% BSA. Sporozoites were then deposited in BSA-coated 96-well plates, centrifuged for 2 min at 100 x g and placed at 37°C in the microscope chamber with 5% $CO_2$. Acquisitions were initiated after 10 min, to allow sporozoite activation, with one image captured every second for a total of 3 min. Images were acquired on an Axio Observer 7 widefield microscope (Zeiss), with ORCA-Fusion Digital CMOS cameras and a Duolink camera adapter for simultaneous two-color acquisition with integrated multi-bandpass emission filter cubes for efficient image acquisition. The microscope was equipped with a 20x objective and ZEN acquisition software. Movies (181 frames per movie) were projected into a single image for each channel and merged using ImageJ. Sporozoites were counted manually based on their movement pattern and divided into four groups: circular gliding, floating, waving or fixed. Rapamycin-exposed and untreated *clamp*cKO sporozoites were differentiated based on their fluorescence (after merge: red for CLAMP-deficient rapamycin-exposed parasites and yellow for control untreated parasites). Three independent motility experiments were performed,

with a total of 1496 and 864 sporozoites analyzed in the untreated and rapamycin-treated conditions, respectively.

## Immunoprecipitation assay and mass spectrometry analysis

Freshly dissected untreated *clamp*cKO sporozoites were lysed on ice for 30 min in a lysis buffer containing 10mM Tris pH 7.5, 150mM NaCl, 0.5 mM EDTA, 0.5% w/v NP40 (Igepal CA-630, Sigma) and protease inhibitors. After centrifugation (15,000 g, 15 min, 4°C), supernatants were collected and incubated with protein G-conjugated sepharose for preclearing overnight. Precleared lysates were subjected to CLAMP-Flag immunoprecipitation using Anti-FLAG M2 Affinity Gel (Sigma) for 2h at 4°C, according to the manufacturer's protocol. PbGFP parasites with untagged proteins were used as a control and treated in the same fashion. After washes, proteins on beads were eluted in 2X Laemmli and denatured (95°C, 5 min). After centrifugation, supernatants were collected for further analysis. Samples were subjected to a short SDS-PAGE migration, and gel pieces were processed for protein trypsin digestion by the DigestProMSi robot (Intavis), as described [10]. Peptide samples were analyzed on a timsTOF PRO mass spectrometer (Bruker) coupled to the nanoElute HPLC, as described [10]. Mascot generic files were processed with X! Tandem pipeline (version 0.2.36) using the PlasmoDB_PB_39_PbergheiANKA database, as described [10]. Data were obtained from 2 independent *clamp*cKO (untreated) and 3 control PbGFP sporozoite lysates, and are provided in **S1 Table**. To confirm the presence of TRAP in CLAMP immunoprecipitates, additional *clamp*cKO (untreated) and PbGFP sporozoite lysates were independently prepared and analyzed by western blotting after anti-Flag IP. The mass spectrometry proteomics data have been deposited to the ProteomeXchange Consortium via the PRIDE [58] partner repository with the dataset identifier PXD037561.

## Western-blot analysis

Rapamycin-exposed and untreated *clamp*cKO sporozoites were collected from the haemolymph of infected mosquitoes at day 16 post-transmission and resuspended in 1X PBS. For analysis of TRAP shedding, microneme secretion was stimulated by incubation for 15 min at 37°C in a buffer containing 1% BSA and 1% ethanol, as described [25]. Pellet and supernatant fractions were then isolated by centrifugation, resuspended in Laemmli buffer and analyzed by SDS-PAGE under non-reducing conditions. Western blotting was performed using rabbit polyclonal antibodies against TRAP [26], 3D11 monoclonal antibody against CSP [59], or M2 monoclonal antibody against Flag (Sigma), and secondary antibodies coupled with Alexa Fluor 680 or 800. Membranes were then analyzed using the InfraRed Odyssey system (Licor). Band intensities were quantified using ImageJ.

## Ultrastructure expansion microscopy

Sporozoites were collected from infected mosquito salivary glands, centrifuged at 3800 g during 4 min at 4°C and resuspended in 1X PBS. Parasites were sedimented on poly-D-lysine coverslips (100 μL/coverslip) during 30 min at room temperature (RT). Samples were then prepared for U-ExM as previously published [27,29] with some modifications. Briefly, coverslips were incubated overnight in a 2% Acrylamide/1.4% Formaldehyde solution at 37°C. Gelation was then performed in 10% ammonium persulfate (APS)/10% Temed in monomer solution (19% Sodium Acrylate; 10% Acrylamide; 0.1% BIS-Acrylamide in PBS) during 1 h at 37°C. Following gelation, denaturation was performed in 200mM SDS, 200mM NaCl and 50mM Tris pH 9.0 during 90 min at 95°C. A first round of expansion was performed by incubating the gels thrice in ultrapure water for 30 min at RT. Gels were then washed in PBS twice for 15 min to remove excess water and blocked with 2% BSA in PBS

for 30 min at RT. Staining was then performed by incubation with primary antibodies diluted at 1/250 in PBS containing 2% BSA at RT overnight with 120–160 rpm shaking. We used antibodies against Flag (clone M2, Sigma F1804), TRAP [26] and AMA1 (1:250, clone 28G2, MRA-897A, Bei Resources). The next day, gels were washed 3 times for 10 min in PBS-Tween 0.1%. Incubation with the secondary antibodies was performed for 3 h at RT with 120–160 rpm shaking, followed by 3 washes of 10 minutes in PBS-Tween 0.1%. Directly after washing, gels were expanded for a second round in ultrapure water for 30 min, thrice. For imaging, 5 mm x 5 mm gel pieces were cut from the expanded gels and mounted between glass slides and Poly-D-Lysine coated coverslips. Acquisitions were made on a Zeiss Axio Imager Z1 fluorescence microscope equipped with a Plan-Apochromat 100x/1.40 Oil DIC M27 objective, using the AxioVision software (Zeiss). Images were processed with ImageJ for adjustment of contrast.

## Statistical analysis

Statistical significance was assessed by two-way ANOVA, ratio paired t tests or Chi-squared test, as indicated in the figure legends. All statistical tests were computed with GraphPad Prism 7 (GraphPad Software). *In vitro* experiments were performed with a minimum of three technical replicates per experiment. Quantitative source data are provided in **S3 Table**.

## Supporting information

**S1 Table. *P. berghei* proteins identified by mass spectrometry after anti-Flag immunoprecipitation.**
(XLSX)

**S2 Table. List of oligonucleotides used in the study.**
(XLSX)

**S3 Table. Quantitative source data and statistical analysis.**
(XLSX)

**S1 Fig. Generation of *P. berghei clamp*cKO parasites. A.** Detailed strategy to insert a LoxN site upstream of *clamp* gene using the P1 construct. Upstream homology regions (5'HR1 and 5'HR2) were inserted in the pUpstream2Lox plasmid on each side of a GFP-2A-hDHFR cassette flanked by two LoxN sites. The P1 construct was transfected into mCherry-expressing PbDiCre parasites. Following parasite transfection and selection with pyrimethamine, mCherry[+]/GFP[+] parasites were sorted by flow cytometry to exclude any residual GFP[-] population. Rapamycin-induced excision lead to removal of the GFP-2A-hDHFR cassette and the retention of a single LoxN site upstream of *clamp*. Genotyping primers and expected PCR fragments are indicated by arrows and lines, respectively. **B.** PCR analysis of genomic DNA isolated from parental PbDiCre and *clamp*-P1 parasites. Confirmation of the predicted recombination events was assessed with primer combinations specific for WT, 5' or 3' integration for the first transfection (T1). Primers used for genotyping are listed in **S2 Table**. **C.** Detailed strategy to insert a LoxN site downstream of *clamp* gene using the P2 construct. Downstream 3' homology regions (3'HR1 and 3'HR2) were inserted in the pDownstream1Lox plasmid on each side of a GFP-2A-hDHFR cassette, flanked on one side by a single LoxN site. A triple Flag epitope tag (3xFlag) was inserted in frame with *clamp* ORF immediately before the STOP codon. In addition, a 559 bp fragment corresponding to the 3' UTR sequence from *P. yoelii clamp* gene was inserted immediately downstream of STOP codon, to allow proper gene expression and avoid spontaneous recombination with the 3' UTR of *P. berghei* clamp, which was used as 3'HR2. The P2 construct was transfected into rapamycin-treated

mCherry<sup>+</sup>/GFP<sup>-</sup> *clamp*-P1 parasites (*clamp*-P1<sup>rapa</sup>). Following parasite transfection and selection with pyrimethamine, mCherry<sup>+</sup>/GFP<sup>+</sup> parasites were sorted by flow cytometry to exclude any residual GFP<sup>-</sup> population, and cloned by limiting dilution and injection into mice, resulting in the final *clamp*cKO parasite line. Exposure of *clamp*cKO parasites to rapamycin leads to excision of *clamp* gene together with the GFP-2A-hDHFR cassette. Genotyping primers and expected PCR fragments are indicated by arrows and lines, respectively. **D.** PCR analysis of genomic DNA isolated from parental PbDiCre, *clamp*-P1 and *clamp*cKO parasites. Confirmation of predicted recombination events was assessed with primer combinations specific for WT, 5' or 3' integration for the second transfection (T2). A band in *clamp*-P1 gDNA for 3' integration is due to the forward primer which can bind to the PbCAM 3' UTR sequence integrated upstream of *clamp* during the first transfection, and is indicated by an asterisk. Primers used for genotyping are listed in **S2 Table**.
(TIF)

**S2 Fig. Analysis of *clamp*cKO parasite development in mosquitoes. A-B.** Fluorescence-based quantification of excised (mCherry<sup>+</sup>GFP<sup>-</sup>) and non-excised (mCherry<sup>+</sup>GFP<sup>+</sup>) sporozoites collected from midguts (A) and salivary glands (B) of female mosquitoes infected with rapamycin-exposed and untreated *clamp*cKO parasites. Results shown are based on observation of at least 200 sporozoites per condition and per experiment (mean +/- SEM of four independent experiments). **C.** Quantification of infected female mosquitoes exhibiting mCherry-labelled pericardial cells 16 days post-infection, based on observation of at least 50 mosquitoes per condition (mean +/- SEM of four independent experiments). Ns, non-significant (Two-tailed ratio paired t test).
(TIF)

**S3 Fig. Analysis of sporozoites by expansion microscopy. A-B.** Salivary gland sporozoites expressing Flag-tagged CLAMP (untreated *clamp*cKO parasites) were examined by expansion microscopy using antibodies against Flag (green) and TRAP (in A, red) or AMA1 (in B, red). Specific accumulation of CLAMP is clearly visible at the apical tip of sporozoites (arrows). Scale bars, 10 μm. **C.** Rapamycin-treated *clamp*cKO parasites were examined by expansion microscopy after labeling with antibodies against Flag (green) and TRAP (red). The absence of signal with anti-Flag antibodies confirms the efficient depletion of CLAMP following rapamycin-induced gene excision. Scale bars, 10 μm.
(TIF)

**S4 Fig. TRAP expression on the surface of salivary gland sporozoites.** Sporozoites were collected from the salivary glands of mosquitoes infected with untreated or rapamycin-exposed *clamp*cKO parasites. Microneme secretion was stimulated by incubation at 37°C in the presence of 1% BSA and 1% ethanol for 15 min. Stimulated and unstimulated sporozoites were then fixed with 4% PFA without permeabilization, and stained with anti-TRAP antibodies (magenta) and the nuclear stain Hoechst 33342 (blue). Untreated parasites express GFP (green) and mCherry (red), while rapamycin-treated parasites express mCherry only. Scale bars, 5 μm.
(TIF)

**S1 Movie. Motility of untreated and rapamycin-exposed *clamp*cKO salivary gland sporozoites.** Untreated (mCherry<sup>+</sup>/GFP<sup>+</sup>, yellow) and rapamycin-treated (mCherry<sup>+</sup>/GFP<sup>-</sup>, red) *clamp*cKO sporozoites were mixed at a 1:1 ratio. Motility was recorded with one frame per second (fps) for 3 min after activation in 3% BSA. Video was edited with 7fps. Only untreated parasites show circular gliding activity.
(MP4)

## Acknowledgments

We thank Maurel Tefit and Thierry Houpert for rearing of mosquitoes, Sylvie Briquet, Aurélie Chauffour and Samhita Das for technical assistance, Freddy Frischknecht and Jessica Kehrer for the kind gift of TRAP antibodies, and Amandine Guérin and Mathieu Brochet for sharing expansion microscopy protocols. The following reagent was obtained through BEI Resources, NIAID, NIH: Monoclonal Anti-Plasmodium Apical Membrane Antigen 1, Clone 28G2 (produced in vitro), MRA-897A, contributed by Alan W. Thomas. This work benefited from equipment and services from the P3S core facility, a platform supported by the Conseil Régional d'Ile-de-France, Sorbonne Université, the National Institute for Health and Medical Research (INSERM) and the Biology, Health and Agronomy Infrastructure (IBiSA), and from the ICM.Quant core facility (Paris Brain Institute), a platform supported through the ANR grants, ANR-10-IAIHU-06 and ANR-11-INBS-0011-NeurATRIS.

## Author Contributions

**Conceptualization:** Manon Loubens, Olivier Silvie.

**Formal analysis:** Manon Loubens, Olivier Silvie.

**Funding acquisition:** Olivier Silvie.

**Investigation:** Manon Loubens, Carine Marinach, Clara-Eva Paquereau, Soumia Hamada, Bénédicte Hoareau-Coudert, David Akbar, Jean-François Franetich, Olivier Silvie.

**Methodology:** Manon Loubens.

**Supervision:** Olivier Silvie.

**Writing – original draft:** Manon Loubens, Olivier Silvie.

**Writing – review & editing:** Manon Loubens, Carine Marinach, Clara-Eva Paquereau, Soumia Hamada, Bénédicte Hoareau-Coudert, David Akbar, Jean-François Franetich, Olivier Silvie.

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
