## [Decision Letter · Decision Letter 0]

13 Oct 2022

Dear Olivier,

Thank you very much for submitting your manuscript "The claudin-like apicomplexan microneme protein is required for gliding motility and infectivity of Plasmodium sporozoites" for consideration at PLOS Pathogens. As with all papers reviewed by the journal, your manuscript was reviewed by members of the editorial board and by several independent reviewers. In light of the reviews (below this email), we would like to invite the resubmission of a significantly-revised version that takes into account the reviewers' comments.

As you will see, the reviewers agreed that the work is of very high quality and that it generates important data regarding the function of CLAMP in sporozoite biology, as well as demonstrating the applicability of the DiCre conditional gene disruption system in insect stages of Plasmodium. However, Reviewer #1 believes that the work produces few new mechanistic insights into the functional interactions between CLAMP and its partner proteins, and specifically how CLAMP modulates TRAP function. The editorial board fully concurs with this view, especially in the context of previous studies on CLAMP. The apparent physical interaction with TRAP and the putative pantothenate transporter PAT (but please see PMID: 34969059 regarding PAT) is intriguing, but to provide the level of new insight to warrant further consideration by PLoS Pathogens it is essential that the authors better elucidate the mechanism(s) underlying the apparent regulation of TRAP secretion by CLAMP. Accordingly, we would be willing to consider a revised form of the manuscript if you can provide such data. As suggested by the Reviewer, this could comprise more detailed analysis of the interaction between the proteins that co-precipitate in your experiments (do they form a multimolecular complex?), as well as information on the subcellular localisation of the partner proteins in sporozoites.

We cannot make any decision about publication until we have seen the revised manuscript and your response to the reviewers' comments. Your revised manuscript is also likely to be sent to reviewers for further evaluation.

Sincerely,

Michael J Blackman

Associate Editor

PLOS Pathogens

Kami Kim

Section Editor

PLOS Pathogens

Kasturi Haldar

Editor-in-Chief

PLOS Pathogens

orcid.org/0000-0001-5065-158X

Michael Malim

Editor-in-Chief

PLOS Pathogens

orcid.org/0000-0002-7699-2064

As you will see, the reviewers agreed that the work is of very high quality and that it generates important data regarding the function of CLAMP in sporozoite biology, as well as demonstrating the applicability of the DiCre conditional gene disruption system in insect stages of Plasmodium. However, Reviewer #1 believes that the work produces few new mechanistic insights into the functional interactions between CLAMP and its partner proteins, and specifically how CLAMP modulates TRAP function. The Editorial Board fully concurs with this view, especially in the context of previous studies on CLAMP. The apparent physical interaction with TRAP and the putative pantothenate transporter PAT (but please see PMID: 34969059 regarding PAT ) is intriguing, but to provide the level of new insight to warrant further consideration by PLoS Pathogens it is essential that the authors better elucidate the mechanism(s) underlying the apparent regulation of TRAP secretion by CLAMP. Accordingly, we would be willing to consider a revised form of the manuscript if you can provide such data. As suggested by the Reviewer, this could comprise more detailed analysis of the interaction between the proteins that co-precipitate in your experiments (do they form a multimolecular complex?), as well as information on the subcellular localisation of the partner proteins in sporozoites.

Reviewer's Responses to Questions

**Part I - Summary**

Reviewer #1: In this paper the authors use their recently established DiCre system to delete specifically in mosquito stages of the Plasmodium parasite the gene encoding CLAMP, a protein previously shown to be important in host cell invasion in Toxoplasma gondii. The authors show that CLAMP is important in salivary gland invasion in mosquitos. Importantly, the authors could immunoprecipitate two other proteins that are essential for salivary gland invasion suggesting that the three proteins built a complex.

The experiments are well conducted and the paper is very well written. The paper is an important contribution on the biological side with the discovery of a new salivary gland invasion factor of Plasmodium and on the technical side by firmly establishing the DiCre system for conditional gene ablation in Plasmodium mosquito stages.

I feel that the connection between CLAMP and the other two proteins is very exciting and should be made much stronger.

Reviewer #2: The work by Loubens and colleagues addresses the role of the claudin-like apicomplexan microneme protein (CLAMP) in the formation and infectivity of Plasmodium sporozoites. Due to the essentiality of clamp in the asexual blood stages, authors resorted to a very elegant and clear conditional genome editing strategy based on the dimerisable Cre recombinase in P. berghei. The authors successfully deleted the clamp gene in P. berghei transmission stages. They then analyzed its impact on sporozoite formation, egress and capacity to infect mosquitoes' salivary glands and the mammalian host hepatocytes. Data shows that clamp- sporozoites are impaired at invading the mosquito salivary glands and hepatocytes. This severe phenotype was associated with significant defects in gliding motility.

Moreover, this work also demonstrates the DiCre system's robustness for conditional genome editing across P. berghei life cycle.

**Part II – Major Issues: Key Experiments Required for Acceptance**

Reviewer #1: More mechanistic insight is needed as to the connection between CLAMP, TRAP and the pantothenate transporter. Are those proteins localizing in the micronemes together? This could be addressed with immuno electron microscopy and/or by performing pulldowns in parasite lines lacking either TRAP or panthothenate transporter.

Reviewer #2: (No Response)

**Part III – Minor Issues: Editorial and Data Presentation Modifications**

Reviewer #1: Likely three of the six proteins found in the pulldown or contaminants (ELF, actin, tubulin) and this should be stated more clearly.

I would suggests to review the way the figures are arranged as follows. Figure 1: the panel C could be placed next to panel A and panel B could be enlarged. Figure 3: maybe the Höchst and merge signals are not needed. Could the images be enlarged and the micronemes pointed at? Figure 4: the legend to panel C could be moved between the two circles to align the panels

Reviewer #2: Figure 2C: I agree with the authors, that the major defect from clamp deletion is seen in the number of sporozoites in the mosquito salivary glands. However, despite failing to invade the salivary glands, there is no accumulation of clampcKO sporozoites in the hemolymph. Could this be explained by the difference seen when comparing the numbers of midgut sporozoites in clampcKO-infected mosquitoes with the non-treated parasites?

Figure 4A and B: Sporozoites used in these assays are from salivary glands? In the text is written this is the case for the gliding motility assays (Fig4C) but not clear for cell traversal and infection experiments. From the numbers used in the assay and the results it seems so but I suggest clarifying.

Figure 1C: Please confirm if only one experiment was performed.

Line 176: Since no additional purification step is performed, I suggest the use of the term “sporozoites were collected” and not “isolated”. Check all the manuscript.

Line 575: Please include the detailed composition of the lysis buffer as this may affect the lysate composition on membrane-bound proteins.

PLOS authors have the option to publish the peer review history of their article (what does this mean?). If published, this will include your full peer review and any attached files.

Reviewer #1: No

Reviewer #2: No
---

## [Decision Letter · Decision Letter 1]

2 Mar 2023

Dear Dr Silvie,

We are pleased to inform you that your manuscript 'The claudin-like apicomplexan microneme protein is required for gliding motility and infectivity of Plasmodium sporozoites' has been provisionally accepted for publication in PLOS Pathogens.

Best regards,

Michael J Blackman

Academic Editor

PLOS Pathogens

Kami Kim

Section Editor

PLOS Pathogens

Kasturi Haldar

Editor-in-Chief

PLOS Pathogens

orcid.org/0000-0001-5065-158X

Michael Malim

Editor-in-Chief

PLOS Pathogens

orcid.org/0000-0002-7699-2064

Please note that both Reviewers have suggested introducing two very minor amendments to the manuscript (1. the addition of quantification of the images in Figure 6; and 2. some mention of caveats regarding sample preparation for the expansion microscopy) that are will improve it even further. A further formal round of revisions is not required for these changes, but it would be appreciated if these very changes could be made to the manuscript during formatting changes before final acceptance.

Reviewer Comments (if any, and for reference):

Reviewer's Responses to Questions

**Part I - Summary**

Reviewer #1: The revision has improved the paper in my view tremendously. Not only do the authors provide addition (if not final) proof of the interactions (and explain it in a fair way in the rebuttal as well as the limitations in the discussions) but most importantly, the provide the first example of expansion microscopy in sporozoites. This alone is an important feat and the data they generate again strengthen the conclusions.

Reviewer #2: (No Response)

**Part II – Major Issues: Key Experiments Required for Acceptance**

Reviewer #1: none

Reviewer #2: (No Response)

**Part III – Minor Issues: Editorial and Data Presentation Modifications**

Reviewer #1: I wonder if the Expansion microscopy images in Figure 6 could be quantified in any meaningful way, e.g. how many vesicles contain TRAP, how many CLAMP? This might sound tedious to the authors but if they would invest a day or so, it might make this figure even more impressive. And as it's the first time Expansion Microscopy was used, it would really nicely show its power. Maybe also consider to put a non-expanded sporozoite right next to the expanded sporozoite to indicate the level of improved resolution.

Reviewer #2: In the revised version, authors used for the first-time expansion microscopy in Plasmodium sporozoites revealing partial co-distribution of CLAMP with TRAP. The use of this technique also showed an accumulation of CLAMP at the apical tip of the sporozoites. Based on this, authors speculate a possible role for CLAMP at controlling also rhoptries discharge. Despite this accumulation is not seen for TRAP or AMA-1, in the absence of imuno-Tem data and on the fact that this accumulation was not, to my knowledge, evident in IFAs I think authors cannot completely discard any possible effect coming from sample preparation. From the M&M section, before being fixed, sporozoites remain 30 min at RT in poly-D-lysine coated coverslips.

PLOS authors have the option to publish the peer review history of their article (what does this mean?). If published, this will include your full peer review and any attached files.

Reviewer #1: No

Reviewer #2: No

---

## [Editor Report · Acceptance letter]

10 Mar 2023

Dear Dr Silvie,

We are delighted to inform you that your manuscript, "The claudin-like apicomplexan microneme protein is required for gliding motility and infectivity of *Plasmodium* sporozoites," has been formally accepted for publication in PLOS Pathogens.

Best regards,

Kasturi Haldar

Editor-in-Chief

PLOS Pathogens

orcid.org/0000-0001-5065-158X

Michael Malim

Editor-in-Chief

PLOS Pathogens

orcid.org/0000-0002-7699-2064